# Inflammatory and Cardiovascular Biomarkers to Monitor Fabry Disease Progression

**DOI:** 10.3390/ijms25116024

**Published:** 2024-05-30

**Authors:** Adrián Alonso-Núñez, Tania Pérez-Márquez, Marta Alves-Villar, Carlos Fernández-Pereira, Julián Fernández-Martín, Alberto Rivera-Gallego, Cristina Melcón-Crespo, Beatriz San Millán-Tejado, Aurora Ruz-Zafra, Remedios Garofano-López, Rosario Sánchez-Martínez, Elena García-Payá, Manuel López-Mendoza, Ignacio Martín-Suárez, Saida Ortolano

**Affiliations:** 1Rare Diseases & Pediatric Medicine Research Group, Galicia Sur Health Research Institute (IIS Galicia Sur), SERGAS-UVIGO, 36312 Vigo, Spain; adrian.alonso@iisgaliciasur.es (A.A.-N.); tania.perez@iisgaliciasur.es (T.P.-M.); marta.alves@iisgaliciasur.es (M.A.-V.); carlosfernandezpereira@gmail.com (C.F.-P.); jorge.julian.fernandez.martin@sergas.es (J.F.-M.); cristina.melcon.crespo@sergas.es (C.M.-C.); beatriz.san.millan.tejado@sergas.es (B.S.M.-T.); 2Internal Medicine Department, SERGAS-Hospital Alvaro Cunqueiro, 36312 Vigo, Spain; alberto.jose.rivera.gallego@sergas.es; 3Pediatrics Department, SERGAS-Hospital Alvaro Cunqueiro, 36312 Vigo, Spain; 4Molecular Medicine PhD Program, University of Santiago de Compostela, 15782 Santiago de Compostela, Spain; 5Internal Medicine Department, Hospital de la Serranía, 29400 Ronda, Spain; dodita85@hotmail.com; 6Nephrology Department, Hospital de Torrecardenas, 04009 Almeria, Spain; dra_garofano@hotmail.com; 7Rare Disease Research Group, Alicante University General Hospital, Alicante Institute for Health and Biomedical Research (ISABIAL), 03010 Alicante, Spain; rosariosanmar@gmail.com (R.S.-M.); garcia_marpayb@gva.es (E.G.-P.); 8Nephrology Department, Hospital Virgen del Rocío, 41013 Sevilla, Spain; lolopezmen@gmail.com; 9Internal Medicine Department, Hospital Universitario Juan Ramón Jiménez, 21005 Huelva, Spain; imsuarez@telefonica.net

**Keywords:** Fabry disease, enzyme replacement therapy, lysosomal disease, inflammation, cardiovascular biomarkers

## Abstract

Fabry disease is an invalidating multisystemic disorder affecting α-Galactosidase, a rate-limiting hydrolase dedicated to lipid catabolism. Non-metabolized substrates, such as Globotriaosylceramide and its derivatives trigger the direct or indirect activation of inflammatory events and endothelial dysfunction. In spite of the efficacy demonstrated by enzyme replacement therapy or pharmacological chaperones in delaying disease progression, few studies have analyzed whether these treatments can improve the pro-inflammatory state of FD patients. Therefore, the aim of this work was to assess cytokines and cardiovascular risk-related proteins detectable in plasma from FD patients, whether treated or not with ERT, to evaluate the reliability of these markers in monitoring disease stage and treatment effects. We identified inflammatory and endothelial dysfunction markers (ADAMTS-13, TNF-α, GDF-15, MIP-1β, VEGFA, MPO, and MIC-1) that cooperate in a common pathway and are increased in FD patients’ plasma samples. As shown by the assessment of these proteins over time, they can help to evaluate the risk of higher severity in FD, as well as ERT effects. Even though the analyzed proteins cannot be considered as proper biomarkers due to their non-specificity to FD, taken together they can provide a signature of reference molecules with prognostic value for early diagnosis, and evaluation of disease progression and treatment efficacy, using blood samples.

## 1. Introduction

Fabry disease (FD, OMIM#301500) is an invalidating multisystemic disorder caused by defects in *GLA* (Xq22, NC_000023.1, mRNA NM_000169.2), a gene encoding lysosomal α-Galactosidase A (α-GalA, EC 3.2.1.22), which is one of the rate-limiting hydrolases in lipid catabolism β. Non-metabolized substrates, such as Globotriaosylceramide (GL3) and its derivatives (e.g., Lyso-GL3), trigger the direct or indirect activation of inflammatory events and endothelial activity [1,2].

GL3 leads the activation of immune cells and the release of inflammatory cytokines, which can contribute to tissue damage and organ dysfunction. Indeed, GL3 stimulation of monocytes and dendritic cells from healthy individuals triggers an overexpression of Tumor Necrosis Factor α (TNF-α), Interleukin-1 β (IL1-β), and IL-6. These markers are constitutively overexpressed in leukocytes from FD patients and their production increases following stimulation of these cells with lipopolysaccharide (LPS) [3].

Furthermore, it has been shown that in FD patients, GL3 is recognized as an antigen by invariant NKT (iNKT) cells, binding Toll-Like receptor 4 (TLR4), and is presented by the class I major histocompatibility complex CD1d receptor. This causes an imbalance in the set of iNKTs with an increase in the population of CD4^-^ and CD8^-^ double negatives. Double-negative iNKTs trigger an inflammatory reaction by secreting interferon-γ and participate in the production of pro-inflammatory cytokines by macrophages and dendritic cells [3,4].

In more general terms, the activation of TLRs has been related to the receptor binding to molecular factors associated with pathogen infection or tissue damage (such as PAMPs and DAMPs), leading to an increase in reactive oxygen species (ROS) and the induction of tissue remodeling and fibrosis [5].

As a result of these events, individuals with FD can experience a wide range of symptoms, including pain and tingling in the hands and feet, skin rashes, hearing loss, gastrointestinal problems, and severe heart and kidney damage, strictly associated with endothelial activation.

Cardiac involvement is the main cause of premature death in FD [6]. Cardiac involvement starts early in patients, sub-clinically progressing until the onset of left ventricular hypertrophy (LVH), suggestive of hypertrophic cardiomyopathy. LVH is the most common feature in FD, which is usually also present in the majority of patients, who present a late onset phenotype, and is correlated to plasma levels of Lyso-GL3 [7,8,9].

Inflammation and immune dysregulation are key secondary mechanisms of cardiac pathology in FD, with cardiac remodeling biomarkers demonstrating potential relations with disease progression, indicating incipient damage before the onset of structural alterations. A proteomic study performed on a French cohort reported the increase in proinflammatory and prothrombotic factors, such as vascular endothelial growth factor A (VEGFA), VEGFC, Vascular Cell Adhesion Molecule 1 (VCAM), and Intracellular Adhesion Factor Molecule (ICAM), in FD [10]. Another recent proteomic study undertaken in a Spanish population also identified coagulation and vascular function markers such as Apolipoprotein A-IV, Apolipoprotein CIII, and fetuin-A as being altered in FD [11].

Moreover, endothelial dysfunction in FD was shown to be related to the activation of Endothelial Nitric Oxide Synthase (eNOS), directly or indirectly induced by GL3, which leads to increased activity of Inducible Nitric Oxide Synthase (iNOS) and Cyclooxygenase 2 (COX-2), with consequent overexpression of cell adhesion molecules such as ICAM-1, VCAM-1, and E-selectin [12].

Despite the fact that the involvement of inflammation and endothelial dysfunction is well known in FD, the key issue of determining whether inflammatory processes and other factors associated with organ damage remit or normalize with the available treatments remains to be clearly demonstrated.

FD has been treated for almost 20 years with enzyme replacement therapy (ERT), based on the intravenous administration of recombinant human α-GalA (agalsidase alfa or agalsidase beta). Both of the available drugs facilitate the removal of deposits from the vascular endothelium and slow the progression of the disease, in addition to improving aspects of patient’s quality of life, such as the reduction in pain crises [13]. Clinical trials have also demonstrated that ERT can remodel LVH, and improve cardiac function and exercise intolerance [14]. Furthermore, recent reports indicate that the expression of cytokines such as Monocyte Chemoattractant Protein-1 (MCP-1) and IL-6 are elevated in FD but reduced following treatment with ERT [15]. A new pegylated enzyme was also recently approved to be used as ERT in FD, having demonstrated a similar efficacy to that of agalsidase beta in clinical trials [16].

An additional option for FD treatment has also been available since 2016 and it consists in oral administration of a small molecule, Migalastat, which acts as a pharmacological chaperone (PC), stabilizing the mutated enzyme [17]; however, this drug is only indicated for patients with specific *GLA* variants. Migalastat was shown to improve the LVH index, but cardiac biomarker assessment studies of chaperone treatment are still lacking.

Further development of drugs for FD, based on different therapeutic strategies, is ongoing and includes substrate reduction therapy and gene therapy [18,19,20].

Currently, the most specific known biomarker in FD is Lyso-GL3, a metabolite of GL3, whose plasma concentration is increased in FD patients and which can reliably confirm the diagnosis of the disorder [21]. However, there is still controversy about whether Lyso-GL3 is really an indicator for monitoring organ damage, progression of the disease, and the effect of the treatment on the inflammatory process and other complementary pathways involved in FD.

An expert consensus document provides key recommendations for the evaluation of relevant biomarkers in clinical practice to meliorate follow-up of FD patients, who usually present a large phenotypic and genotypic spectrum [22]. The document includes Lyso-GL3 assessment as well as the evaluation of relevant renal and cardiac function indicators (i.e., e-GFR, troponin, NT-ProBNP albuminuria serum creatinine, cardiac MRI, etc.), although it is still unclear how treatments may on these biomarkers to improve patients’ outcomes. Hypothesis-driven exploration of additional biological pathways should help to unravel new biological signatures for optimized diagnosis, management, and therapeutic intervention. Inflammation-related pathways seem to be interesting to explore, as supported by studies on the improvement of treatment efficacy, when combining anti-inflammatory agents with pharmacological chaperone therapy [23,24].

Based on these premises, the aim of this work was to assess inflammatory and endothelial function-related factors in samples from FD patients, treated or not with ERT, to compare their values with those expressed in healthy subjects, and evaluate whether these proteins have prognostic value in the follow-up of FD progression and treatment efficacy.

## 2. Results

### 2.1. Description of Patient Cohorts

The study included 36 Fabry patients, consisting of 17 males and 19 females, with a mean age of 43.25 ± 13.55 years and an age range of 7–66 years. Patients’ features are summarized in Table 1. The diagnosis of Fabry disease was previously established through a medical examination, and confirmed with reduced enzymatic activity and genetic testing. Among the patients, 20 were patients with a classical FD phenotype, 15 presented variants related to late onset FD, and, for 1 patient, the genetic mutation was not disclosed by the physician. We excluded from the study the subjects who presented *GLA* variants of unknown significance or variants related to pseudo-deficiency (e.g., p.Arg188Cys, p.Ala137Tre, p.Asp313Tyr). The majority of the naïve male patients (N = 4 out of 5) were late onset FD patients.

Among the participants, 22 were already under treatment with ERT when they enrolled in the study; therefore the starting point of the analysis (Ti) did not usually correspond with the onset of the treatment. Specifically, 10 patients were receiving agalsidase alfa and 12 patients were receiving agalsidase beta. ERT had been conducted in this cohort for at least for three years before enrollment in the study, except for patient F1 and patient F35. Patient F1 was a naïve female patient, who started to receive ERT right after the first blood extraction at enrolment (Ti). Another naïve male patient started to receive PC 3 months after recruitment.

Control subjects (N = 16) were recruited in primary care medical centers or among volunteers in our research institute. In the control subset, we excluded individuals with chronic inflammatory diseases or oncologic history. Enrolled controls were 9 females and 7 males and presented a mean age of 35.63 ± 8.25 (males = 32.29 ± 7.78 and females = 38.22 ± 8.04 years). Relevant features for this cohort are summarized in Appendix A.

### 2.2. Concentration of Circulating Inflammation Markers

Biomarkers related to inflammation were initially assessed by Luminex-multiplex ELISA in plasma from FD patients treated with ERT (N = 18; 10 females and 8 males) or non-treated (N = 6; 5 females and 1 male), and concentration values were compared with those obtained in healthy subjects (N = 10). In this assay, we assessed cytokines such as IL-1α, IL-1β, IL-4, IL-5, IL-6, IL-9, IL-10, IL-12p40, IL-12p70, IL13, IL-17a, Interferon-γ, MCP-1, Macrophage inflammatory protein (MIP)-1α, MIP-1β, TNF-α, and VEGFA (Figure 1 and Appendix A).

The concentration of 11 cytokines was below or close to the detection limits of the assay with the applied protocol. Proteins like IL-12p70 (Z score average: control: −0.18 ± 0.22 FD naïve: −0.43 ± 0.07 FD ERT: 0.06 ± 0.29) and IL-10 (Z score average: control: 0.09 ± 0.34 FD naïve: 0.25 ± 0 FD ERT: −0.10 ± 0.30) were detectable, and therefore probably increased, in some of the FD patients of the cohort, while IL-17A (Z score average: control: 1.15 ± 0.30 FD naïve: −0.23 ± 0.26 FD ERT: −0.56 ± 0.11) was more easily detectable in controls (Appendix A). Average Z-score values for this panel are reported in Appendix A.

Due to the low concentration of many cytokines in the collected samples, we decided to carry out further validation analysis only in the cytokines that were detectable in most of the subjects and overexpressed in FD naïve or ERT-treated patients compared to healthy subjects, such as: TNF-α, VEGFA, MCP-1, and MIP-1β (average values of Z-scores for the selected markers are TNF-α: control: −0.50 ± 0.09 FD naïve: −0.32 ± 0.49, FD ERT: 0.38 ± 0.25; VEGFA: control: −0.38 ± 0.11 FD naïve: −0.05 ± 0.47, FD ERT: 0.04 ± 0.30; MCP-1: control: −0.34 ± 0.14 FD naïve: −0.02 ± 0.13 FD ERT: 0.19 ± 0.31; and MIP-1β: control: −0.39 ± 0.01 FD naïve: 0.26 ± 0.56, FD ERT: 0.13 ± 0.26). For this purpose, we performed conventional ELISA analysis in a greater number of patients (controls, N = 16; naïve FD patients, N = 14; ERT FD patients, N = 22).

Mean plasmatic TNF-α concentration (Figure 2) was significantly increased in plasma from FD patients compared to healthy controls (6.68 ± 0.67 pg/mL in controls and 12.16 ± 1.47 pg/mL in FD patients: in naïve 11.79 ± 2.98 pg/mL and in ERT-treated 12.41 ± 1.47 pg/mL). Sex stratification of the study subjects did not show significant differences in the concentration of TNF-α in the analyzed cohorts, although average values were higher in FD patients of both sexes compared to healthy controls (control males: 7.23 ± 0.88 pg/mL; FD males: 15.25 ± 2.70 pg/mL; control females: 6.66 ± 0.98 pg/mL; FD females: 9.23 ± 0.94 pg/mL).

Measurements of TNF-α plasmatic concentration in samples collected 6 or 12 months after the onset of the study (Ti) demonstrated that TNF-α concentration is significantly decreased in plasma from ERT-treated males and females, after 12 months, in comparison with Ti and naïve patients’ values. Details on the statistical analysis process that was applied and the *p*-value tables are presented in Appendix A.

Of note, TNF-α concentration in plasma was also decreased following 12 months in patient F1, who started ERT at the beginning of the protocol (Ti= 7.78 pg/mL, after 6 months 8.36 pg/mL and 3.98 pg/mL after 12 months) and in patient F35, who started chaperone therapy 3 months after Ti (Ti = 9.6 pg/mL, after nine months of treatment 3.2 pg/mL). A representation of the plasma TNF-α concentration time course in single patients is included in Appendix A.

Mean plasmatic VEGFA concentration (Figure 3) was significantly increased in plasma from FD patients compared to healthy controls (2.97 ± 0.79 pg/mL in controls and 8.63 ± 1.83 pg/mL in FD patients: in naïve 6.64 ± 2.47 pg/mL and in ERT-treated 10.42 ± 2.56 pg/mL), and the difference was also significant when comparing the male FD patient group with male healthy controls. We could not find significant differences or a clear trend in the evolution of VEGFA concentration over time in the different groups in both male and female cohorts (Appendix A).

Mean plasmatic MCP-1 concentration (Figure 4) was significantly increased in plasma from FD patients compared to healthy controls (46.69 ± 3.96 ng/mL in controls and 60.08 ± 9.06 ng/mL in FD patients: in naïve 53.23 ± 4.70 ng/mL and in ERT-treated 65.22 ± 18.78 ng/mL).

In the treatment groups, we detected stable or increased concentrations of MCP-1 in the patient cohort, and a significant decrease in the protein concentration in the patients treated with ERT 12 months after Ti. No significant differences were detected in the ERT patients when analyzing data of male and female groups separately. MCP-1 concentration was also decreased after 12 months in plasma from patient F1, who started treatment at Ti (Ti = 59.70 ng/mL, after 6 months 59.85 ng/mL and 26.69 ng/mL after 12 months). Nonetheless, we saw an increase in MCP-1 concentration in plasma from two ERT treated patients (F12 and F26), who were followed for only six months (Appendix A).

Mean plasmatic MIP-1β concentration (Appendix A) was not significantly increased in plasma from FD patients compared to healthy controls (13.08 ± 2.58 pg/mL in controls and 14.36 ± 22.99 pg/mL in FD patients: in naïve 11.14 ± 1.78 pg/mL and in ERT-treated 18.44 ± 37.91 pg/mL), and the obtained values were very close to the detection limit of the assay in most of the samples.

### 2.3. Concentration of Circulating Cardiovascular Markers

A panel of cardiovascular risk related biomarkers were also assessed using the ELISA-Luminex cardiovascular approach in plasma from FD patients treated or not with ERT (controls N = 10; FD naïve N = 6 and FD-ERT patients N = 18). The panel detected A disintegrin and metalloproteinase with thrombospondin type 1 repeats, member 13 (ADAMTS-13), Growth differentiation factor 15 (GDF-15), Myoglobin, s-ICAM, s-VCAM, P-selectin, Lipocalin-2, Metalloproteins myeloperoxidase (MPO), D-dimer, and serum amyloid (SAA).

Most of the analyzed cardiovascular biomarkers were increased in FD patients in comparison with healthy controls (Figure 5, Appendix A, Appendix A).

However, the mean plasma concentration of MPO, ADAMTS-13, and GDF-15 (average values of the Z-score for the selected markers are ADAMTS-13: control: 0.05 ± 0.27, FD naïve: 1.21 ± 0.62, FD ERT: −0.43 ± 0.03; GDF-15: control: −0.43 ± 0.04 FD naïve: 0.36 ± 0.62, FD ERT: 0.12 ± 0.07; and MPO: control: −0.70 ± 0.05 FD naïve: 0.43 ± 0.53 FD ERT: 0.25 ± 0.06) was lower in the ERT-treated cohort compared to non-treated patients; therefore, we selected these markers for further validation in plasma samples from additional FD patients and healthy controls by conventional ELISA or a multiplex assay. With these samples, we also assessed the evolution of cardiovascular risk-related proteins during the 12 months of the study.

In the validation cohorts, mean plasmatic ADAMTS-13 concentration (Figure 6) was not significantly different in plasma from FD patients compared to healthy controls, although we could appreciate from mean values (513.06 ± 37.54 ng/mL in controls and 672.14 ± 51.88 ng/mL in FD patients: in naïve 690.21 ± 65.40 ng/mL and in ERT-treated 658.34 ± 70.24 ng/mL) that the concentration of ADAMTS-13 tended to be higher in both naïve and ERT-treated patient cohorts.

Moreover, analyzing the follow-up samples, we detected that the concentration of ADAMTS-13 remained stable in naïve patients during 12 months of study, while it progressively and significantly decreased in ERT-treated patients after 12 months. Looking at differences between sexes, we detected that plasma levels of ADAMTS-13 were significantly decreasing following 12 months of the protocol in female patients. Remarkably, the concentration of ADAMTS-13 was also decreased over time in the female patient (F1) who started the treatment right after Ti blood extraction (Ti = 548.66 mg/mL, after 6 months: 543.72 mg/mL, after 12 months: 282.52 mg/mL). In patient F35, treated with the oral therapy three months after Ti, ADAMTS-13 concentration was also decreased at the 12 months follow-up (Ti = 596.76 ± 13.40, after 9 months: 249.11 ± 2.50) (Appendix A).

Similar to ADAMTS-13 results, the mean plasmatic concentration of GDF-15 (Figure 7) was higher in FD patients compared to controls, although the differences were not significant among the groups (1.95 ± 0.38 ng/mL in controls and 2.90 ± 0.48 ng/mL in FD patients: in naïve 2.46 ± 0.56 ng/mL and in ERT-treated 3.19 ± 0.61 ng/mL).

The evolution of GDF-15 over time confirmed that the concentration of the protein stayed stable or increased in naïve patients, while it tended to decrease in ERT-treated patients, especially in females, including patient F1 (Ti = 1.70 ng/mL, after 6 months 1.47 ng/mL and 0.52 ng/mL after 12 months) (Appendix A).

Mean plasmatic MPO concentration (Appendix A) was increased in FD patients compared to healthy controls (483.66 ± 86.30 ng/mL in controls and 1461.88 ± 214.64 ng/mL in FD patients: in naïve 1353.01 ± 365.65 ng/mL and in ERT-treated 1539.64 ± 262.42 ng/mL) and was significantly different in male patients versus female FD patients. The values of this biomarker over time were not significantly different in the analyzed cohort, and also did not show a clear decreasing trend in ERT-treated patients.

### 2.4. α-GalA Activity in Plasma

α-GalA activity was assessed in plasma of the studied subjects at enrollment (Ti) to evaluate possible correlations with the other evaluated biomarkers. Plasma levels of α-GalA activity reported in the literature are generally lower than those detected in DBS or leukocytes, indicating a low level of circulating enzyme also in control subjects or heterozygous female patients [25]. As expected, the activity of the enzyme was significantly higher in controls (N = 16) compared to FD naïve patients (N = 14) (Figure 8A), and female patients presented slightly higher activity values than male patients (Figure 8B). In patients treated with ERT at Ti (N = 22) plasmatic activity levels were not significantly different from controls; however, ERT-treated patients presented a slight increase in the α-GalA activity compared to naïve patients, even though the samples were collected before the administration of the drug infusion.

### 2.5. Concentration of Circulating Anti-α-GalA-IgG Antibodies

Concentration of anti-α-GalA IgG antibodies was determined in plasma samples (Figure 9). The threshold for anti-α-GalA IgG antibody detection (110 μg/mL) was established above the concentration levels detected for controls and naïve patients. The measurements were carried out in plasma samples instead of serum samples because serum samples were only available for a few subjects. The sensibility of the assay in plasma was reduced, but all the subjects identified as positive were confirmed with a clinically validated assay (performed externally at Labcorp).

Three patients presented an anti-α-GalA IgG concentration in plasma above the threshold and one of them had values close to this limit (LSD-01). Specifically, in one patient (F10), the concentration of antibodies decreased over time and was below the threshold in the 12-month follow-up sample, while in the remaining two patients, the plasmatic concentration of antibodies increased over time (F7 and F21) when continuing the administration of enzyme replacement therapy. Of note, patient F7 was a female patient who is homozygous for the mutation p.Gln279Arg due to consanguinity of her parents, so her phenotype was, in many aspects, close to that of a male patient [26]. The LSD-01 patient, who presented anti-α-GalA IgG values close to the defined threshold, underwent kidney transplant prior to inclusion in this study.

α-GalA activity and Lyso-GL3 concentration in plasma were not affected in samples from patients F7 and F10, while these values were inversely correlated in patient F21 (activity: Ti, 0.175 ± 0.04 μg/mL; T6, 0.140 ± 0.00 μg/mL; T12, 0.210 ± 0.01 μg/mL, Lyso-GL3: Ti, 15.74 ng/mL; T6, 11.60 ng/mL; T12, 16.92 ng/mL).

### 2.6. Lyso-GL3 Concentration in Plasma

Concentration of Lyso-GL3 in plasma was significantly increased in plasma from FD patients independently of the treatment, in comparison with samples from control individuals (Figure 10). A non-significant decrease in Lyso-GL3 was observed over 1 year of follow-up in patients treated with ERT. A marked reduction in Lyso-GL3 concentration was observed in samples from patient F1, who started the treatment at Ti (Ti, 1.16 ng/mL; T6, 0.67 ng/mL; T12, 0.16 ng/mL).

### 2.7. Correlation of Analyzed Biomarker Concentrations

The correlation among the measured biomarkers was evaluated to establish possible interdependence between variables. By calculating the Spearman coefficient, we found significative correlations between most of the variables, considering data from all groups at the starting point (Ti, Appendix A).

As expected, most of the inflammatory cytokines and cardiovascular variables were interrelated. For example, plasma levels of ADAMTS-13 were obviously correlated with D-Dimer, but also with GDF-15, MPO, and MIP-1β, while TNF-α was correlated with MIP-1β expression and GDF-15 showed a significative correlation with MIP-1β and D-dimer.

Although most of the evaluated proteins usually correlate with aging, in our cohort the only variables which presented a significant correlation with aging were Lyso-GL3 and VEGFA. These findings suggest that the majority of the significant correlations that we found among the assessed variables are unrelated to the difference in the mean age between the control and patient groups, and are most likely due to the direct or indirect effect of the disease.

Looking at the segregation of the data in terms of sex, Lyso-GL3 correlates with plasma activity and MPO in males but not in females. The sex-related differences between variables for the other analyzed biomarkers were limited to ADAMTS-13, which showed a positive correlation with MIP-1β only in males, and with D-dimer and GDF-15 in both sexes. Moreover, in females, VEGFA and MPO correlated with age and GDF-15 with VEGFA.

A second correlation analysis was carried out for all the variables in the ERT groups at the beginning of the study and after 6 or 12 months of follow-up. At the starting point, significative correlations between variables were similar in the ERT group compared to the whole cohort of patients. However, in the cohort of ERT-treatment patients after 1 year of follow-up, the plasma concentrations of MPO and VEGFA were also correlated with the concentration of Lyso-GL3 (Appendix A). These findings support the dependence between the indicated inflammation and cardiovascular risk-related proteins with specific FD biomarkers.

## 3. Discussion

The present work reports the results of an observational study that assessed variables linked to inflammation and cardiovascular risk in cohorts of FD patients and healthy controls, who were followed-up for 12 months.

We identified inflammatory and endothelial function-related factors directly or indirectly involved in FD pathophysiology, which possibly correlate with disease progression and treatment effects. Specifically, we demonstrated that plasmatic levels of TNF-α, VEGFA, and MCP-1 are significantly increased in FD patients and that TNF-α, MCP-1, and ADAMTS-13 concentrations showed a significant decrease over time in ERT-treated patients. Moreover, the levels of ADAMTS-13, MIP-1β, GDF-15, and MPO also tended to increase in FD patients in comparison with healthy controls, even if the difference did not reach significant values with the number of analyzed patients. Additionally, GDF-15 and MIP-1β tended to decrease after 12 months of follow-up in the ERT cohort.

Inflammatory markers were assessed in FD in previous studies, although these projects were usually developed in cohorts that did not discriminate between treated and untreated patients, or they reported the results of indirect measurements performed after cell stimulation. As an example, TNF-α levels were shown to be increased in culture media from FD patients’ cultured lymphocytes upon stimulation of TLR4 by GL3 stimulation [3,27].

In our study, we performed direct measurements in plasma samples, and we were able to compare the evolution of biomarker concentrations in naïve patients with patients treated with ERT, in which we could demonstrate a positive effect of the treatment in the modulation of specific inflammatory or cardiovascular risk factors. The effect of the ERT treatment in normalizing levels of TNF-α, MIP-1β, and ADAMTS-13 was confirmed by the analysis of these proteins in plasma samples obtained pre- and post-ERT onset in patient F1, who was treated with ERT and F35, and treated with PC.

Our data support the results obtained in a cohort of patients from Taiwan (patients with IVS4 + 919G > A in *GLA*), in which a significant increase in plasma concentration of TNF-α was shown in FD patients who presented with cardiomyopathy. In this study, it was also demonstrated that TNF-α concentration was reduced upon treatment with ERT [15].

The majority of FD patients in our cohort also presented with cardiomyopathy since they were affected with late onset cardiac mutations (e.g., p.Ser238Asn, p.Pro205Ser) or classical FD mutations with severe cardiac involvement (p.Gln279Arg). All these patients presented LVH, although specific LVH indexes and values of troponin were not available for all the patients in the cohort to perform a direct correlation.

Neto et al. [28] also described an increased concentration of TNF-α in a female FD cohort of patients treated with ERT, who likewise presented elevated serum levels of IL-6. In this study, the serum IL-6 and TNF-α levels were correlated with the MSSI scores, reflecting greater disease burden in patients with high cytokine levels. Our study confirms that levels of TNF-α are elevated in naïve patients and decrease over time in ERT-treated male and female patients.

Our data also show that plasma levels of VEGFA, a signaling protein involved in angiogenesis and increased vascular permeability, were increased in FD patients, and especially in male patients. The effects of VEGFA were studied in vascular endothelial cells, although this protein also has effects on other cell types, such as macrophages, neurons, and renal mesangial cells. Since VEGFA increases upon cell stimulation with hydrogen peroxide, its increase in FD could be physiologically related with an increase in oxidative stress [29]. Indeed, VEGFA was shown to be significantly high in FD patients and associated with other characteristic signs of the disease such as angiokeratomas, sweating abnormalities, and Fabry Facies [30,31].

These data are in accordance with the results obtained by a proteomic approach performed on a French cohort of FD, in which four differential proteins were identified as being involved in FD physiopathology. Specifically, these proteins were biomarkers mainly involved in inflammatory and angiogenesis processes, such as fibroblast growth factor 2 (FGF2), VEGFA, VEGFC, and IL-7 [10].

The effect of ERT on inflammation markers was also previously assessed in a cohort of patients with or without LVH, presenting the variant IVS4 + 919G > A in *GLA*, which is related to a predominantly cardiac phenotype [15]. The study concluded that ERT improved Fabry cardiomyopathy (elevated left ventricular mass index and interventricular septal thickness at diastole) and simultaneously determined a decrease in inflammatory markers such as the cytokines MCP-1 and IL-6.

MCP-1 was also significantly increased in FD patients enrolled in this study compared to healthy controls. In the treatment groups, we detected stable or increased concentrations of MCP-1 in the naïve patient cohort, and a significant decrease in the protein concentration in the patients treated with ERT up to 12 months after Ti.

Like inflammatory markers, cardiovascular and endothelial-related risk factors were also associated with FD, and can help to monitor cardiovascular involvement in these patient [32].

A very recent proteomic study undertaken in a Spanish population detected the differential expression in FD patients of multiple proteins involved in the involvement inflammation, heme and hemoglobin metabolism, oxidative stress, coagulation, complement cascade, glucose and lipid metabolism, and glycocalyx formation, some of which were differentially expressed in the two sexes. The main finding of this work points to Apolipoprotein A-IV, a protein involved in platelet aggregation, as a sensitive marker effective at evaluating chronic kidney disease in FD [11].

We demonstrated for the first time that ADAMTS-13, a disintegrine-like metalloproteinase, is overexpressed in FD. ADAMTS-13 cleaves large multimers of Von Willebrand factor to inhibit pathologic platelet activation. Deficiency of ADAMTS-13 has been related to thrombotic thrombocytopenic purpura and to an increased proportion of thrombotic events when Von Willebrand factors or D-Dimer are increased.

The increase in the concentration of this metaloproteinase could be related to a response to the increase in the Von Willebrand factor, which is the substrate of the enzyme (not measured), and D-Dimer, a prothrombotic protein, which strictly correlates with Von Willebrand factor and ADAMTS-13 in different pathologies. In this study, D-Dimer concentration was present in the multiplex assay and therefore evaluated in this cohort, where it was increased in FD patients (Appendix A), although this analyte should be evaluated in serum and not in plasma, according to the manufacturer’s instructions. It is therefore possible that ADAMTS-13 concentration is upregulated in FD to compensate for an increase in Von Willebrand factor or D-dimer. This scenario could also explain the low frequency of thrombotic events that occur in FD patients, which is atypical in subjects with endothelial dysfunction.

On the other hand, ADAMTS-13 presented increased levels in patients with mild and severe renal dysfunction, which also correlates with an increase in D-dimer and Von Willebrand factor [33].

These findings suggest that ADAMTS-13 can be a useful biomarker for the follow-up of FD to follow-up direct or indirect effects of the disease on both cardiovascular and nephrological functions.

GDF-15 was also increased in plasma from the FD patients of our cohort compared to healthy subjects, and showed the same trend as ADAMTS-13, decreasing over time in patients treated with ERT, although these results need to be confirmed in wider cohorts.

GDF-15, a divergent member of the transforming growth factor β superfamily, was also postulated as a potential biomarker to follow-up cardiac and renal involvement in FD [30]. GDF-15 and syndecan-1 were associated with cardiac and renal involvement in classic FD patients on ERT (N = 52). In particular, GDF-15 showed a direct correlation with interventricular septal thickness and estimated glomerular filtration rate. Serum GDF-15 levels were significantly higher in patients with cardiomyopathy, as well in those subjects who presented with both nephropathy and cardiomyopathy, compared to subjects without these comorbidities. The physiological expression of this compound is barely detectable in most tissues, but it is often induced under stress conditions. Highly elevated GDF-15 levels are mostly linked to inflammation, myocardial ischemia, renal pathology, cancer, or age-related frailty [34,35].

Additionally, increased MPO concentration was previously observed in patients with FD, suggesting this could predict FD-associated vasculopathy, since MPO is a peroxidase enzyme secreted by neutrophils during degranulation [36]. Indeed, registry data indicate a high prevalence of risk factors for coronary artery disease in FD that may accelerate conventional atherosclerosis [37,38]. In our cohorts, MPO levels in plasma were increased in FD patients compared to healthy controls and were significantly different in male patients versus female FD patients.

Assessments of characteristic variables related to FD, such as measurement of α-GalA activity, Lyso-GL3 concentration monitoring, and anti-drug IgG antibody detection, were also performed directly in plasma from patients of the studied cohorts to analyze the correlation with the detected biomarkers. We confirmed that Lyso-GL3 concentration tends to increase in non-treated patients and decrease or stay stable in ERT-treated patients, while enzymatic activity does not present a relevant concentration difference in the ERT cohort compared to the naïve cohort, since blood extraction was always carried out before applying the treatment. We also detected that 3 out of 22 patients treated with ERT expressed antibodies against agalsidase. Among these patients, we only detected an increase over time in the plasmatic concentration of Lyso-GL3 and other biomarkers (MPO and VEGFA) in one of the subjects (F21), who could possibly be producing anti α-GalA neutralizing antibodies.

Altered plasma concentration of the analyzed biomarkers confirmed that impaired GL3 catabolization facilitates the activation of endothelial cells and oxidative stress, which are key to triggering a cascade of inflammatory events leading to cardiac and renal disfunction. Statistical analysis of correlations between the analyzed compounds supports the involvement of the examined markers in common pathways, and suggests that these may help in predicting the evolution of FD. By analyzing correlations between variables, we have shown that most of the analyzed proteins are correlated independently of age and that, after 1 year follow-up, the plasmatic concentration of different variables (i.e., MPO, VEGFA) is also correlated with Lyso-GL3 concentration, showing a direct relationship with FD.

We have to point out that this study is subject to some limitations that can be overcome in future by studying a bigger cohort of patients. Indeed, the study included a relatively low number of patients (naïve: N = 14; ERT: N = 22) to show solid statistical significance considering the heterogeneity of the phenotypes of the enrolled patients (classical and late onset with different genetic variants) and the loss of patients in the follow-up groups. However, as discussed, most of the possible biomarkers that were identified in this study were also described in FD cohorts studied by other researchers, supporting the relevance of our findings and the reliability of our methodology. The analysis of a larger number of patients will be required to confirm these data and better stratify patients in terms of other variables, such as the sex or the genotype. Additional studies will also be useful to confirm if the normalization of inflammatory or cardiovascular markers over time is persistent and determined by ERT itself or by a combination of co-adjuvant treatments.

Another important limitation of our study is that the majority of the patients included in the ERT cohort were under treatment for over 3 years, and therefore they were most likely stabilized at the time they entered the study. Variation in biomarker concentrations in these conditions is obviously of smaller magnitude and requires longer follow-up times; indeed, we see a clear tendency of the proposed biomarkers to decrease in the two patients who were naïve at the beginning of the study, and respectively entered ERT (F1) or PC (F35) treatment during the follow-up of the cohort.

Finally, we should point out that the assessment of clinical outcomes that correlate with the biochemical variables would have given more strength to our results, but unfortunately the Mainz Severity Score Index and other clinical variables (e.g., eGFR, troponin, etc.) were not available for the whole cohort, since this was an observational study without intervention in the clinical practice.

Overall, our results suggest that regular monitoring of inflammation and endothelial activation factors is essential in the management of individuals with FD, who present altered levels of these biomarkers, and may help to facilitate prompt decision making at the onset of the treatment or the evolution of the patients to prevent cardiovascular and renal complications.

## 4. Materials and Methods

### 4.1. Patient Recruitment and Sample Processing

Patients were recruited from 6 Spanish centers after signing informed consent. This study was approved by the Galician Ethics Committee of Clinical Investigations with medicaments (CeimG #2018-445, #2019/496 and #2020/419). The included subjects were patients with FD and control individuals of both sexes aged between 7 and 66 years, who signed the informed consent form for the study. Individuals who presented variants of unknown significance in *GLA* were excluded. Among healthy volunteers (control cohort), subjects with a clinical history of cancer, autoimmune diseases, or chronic infectious diseases were also excluded. For all groups, eligible subjects who did not provide consent were excluded.

Following these criteria, 16 healthy control subjects, 14 non-treated patients with FD (FD naïve), and 22 subjects treated with ERT were enrolled. Among the ERT patients, 10 were treated with agalsidase alfa (Replagal™, Takeda Pharmaceutical, Boston, MA, USA) and 12 were treated with agalsidase beta (Fabrazyme^®^, Sanofi-Genzyme, Paris, France). In the cohort of the naïve patients, a female subject switched to ERT, and a male subject switched to PC.

Peripheral blood was collected in Na^+^-Heparin tubes (30 mL) at the onset of the study (Ti) and after 6 or 12 months follow-up. Of note, the start point of the study for treated patients did not correspond with the onset of the treatment in the majority of the recruited patients. Whole blood was diluted 1:2 with PBS and Peripheral Blood Mononucleated cells (PBMCs), and plasma samples were collected after gradient separation with Ficoll Hypaque (#F5415 Sigma-Aldrich-Merck, Darmstadt, Germany). Samples were frozen at -80 ºC before the analysis (cells were preserved with freezing medium #C6164, Sigma-Aldrich-Merck, Darmstadt, Germany).

### 4.2. α-GalA Activity

α-GalA activity was detected in plasma according to a previously described fluorometric method [39]. Briefly, α-GalA mediates the hydrolysis of the substrate 4-methylumbelliferyl-α-D-galactopyranoside (#M7633, Sigma-Aldrich-Merck, Darmstadt, Germany), releasing the 4-methylumbelliferone (4-MU) fluorescent product. Nanomoles of hydrolyzed substrate are calculated from quantified fluorescence relative to a 12-point 4-MU (#1381 Sigma-Aldrich-Merck) in the range 250–0.24 µM. Plasma samples were diluted in 0.15 M Phosphate-Citrate buffer at pH 4.2 and were mixed with substrate at the final concentration of 1.5 mM, in the presence of 200 mM N-acetyl-D-galactosamine (#A2795, Sigma-Aldrich-Merck, Darmstadt, Germany). The microplate was incubated at 37 °C for 30 min; at this point, the stopping solution (0.5 M Sodium Carbonate/Bicarbonate Buffer, pH 10.5) was added to wells to halt the reaction. Fluorescence was read using a FLUOstar^®^ Omega Plate Reader (BMG LABTECH, Offenburg, Germany) at 360 nm excitation and 450 nm emission wavelengths. α-GalA specific activity was expressed as velocity of substrate cleavage (nmol/mL·h). α-Gal activity was also measured in dried blood spots (DBSs) at external labs for diagnostic purposes prior to this study. Activity in DBS is reported in Table 1.

### 4.3. Lyso-GL3 Concentration

Lyso-GL3 concentration was measured by LC-MS in 10 µL of plasma at ARCHIMED Life Science GmbH (Vienna, Austria), using the method described by Nowak et al. [40]. Lyso-GL3 concentration is reported in ng/mL.

### 4.4. Assessment of Inflammatory and Cardiovascular Biomarkers in Plasma

A total of 17 inflammatory and 9 cardiovascular biomarkers were assessed in plasma samples at baseline by MILLIPLEX^®^ Human Magnetic Bead Panels (ThermoScientific, Waltham, MA, USA), using Luminex technology (Millipore kits #HCYTOMAG-60K (Sigma-Aldrich-Merck, Darmstadt, Germany) for cytokines and chemokines, and the HCVD2MAG for cardiovascular markers). Measurements and analysis were performed at the Rafer facility, Zaragoza, Spain.

The plasma levels of selected inflammatory and cardiovascular biomarkers were subsequently validated in our cohort using specific ELISA kits, which detect the human isoform of these proteins (ADAMTS-13: #Ab234559, Abcam; GDF-15: #DY957, MCP-1: #DY279-05; Macrophage inflammatory protein-1β, MIP-1β: #DY271-05; Metalloproteins myeloperoxidase, MPO: #DY3174; TNF-α: # DY210-05, and VEGFA: #DY293B-05, R&D Systems, Minneapolis, MN, USA). Briefly, microwell plates were coated with 100 µL of the respective kit antigen at concentrations ranging from 1 to 4 µg/mL. The plates were incubated overnight at room temperature. After incubation, the wells were washed three times using Wash Buffer (0.05% Tween 20 in PBS, pH 7.4). The plates were blocked using Reagent Diluent (1% BSA in PBS, pH 7.4). Plates were incubated at room temperature for at least 1 h. Another washing step was performed. Samples and standards (duplicates), diluted in reagent diluent, were added, and incubated for 2 h at room temperature. After incubation, another washing step was performed. Each antibody was added at 100 µL (12.5–200 ng/mL) as the primary detecting antibody. The plates were then incubated at room temperature for 2 h. Following the incubation, another washing step was performed. Then, 100 µL of a 1:40 dilution of Streptavidin-HRP was added to each well and incubated for 20 min at room temperature in the dark. After a final series of washing steps, 100 µL of substrate solution (1-Step Ultra TMB-ELISA #34029, ThermoScientific, Waltham, MA, USA) was added to each microwell. To stop the reaction, 50 µL of 2N H2SO4 was added, and the absorbance was read at 450–570 nm in a FLUOstar^®^ Omega Plate Reader (BMG LABTECH, Offenburg, Germany). In the case of TGF-β, prior to sample addition, a supplementary activation step was performed to convert latent TGF-β1 to immunoreactive TGF-β1, by treatment with 1N HCl (10 min at r.t.). The acidified samples were neutralized with 1.2 N NaOH/0.5 M HEPES. The concentration of each analyte was calculated from a standard curve (absorbance versus concentration) of the corresponding protein at a known concentration and expressed as ng/mL.

### 4.5. Anti-α-GalA IgG

The presence of anti α-GalA IgG antibodies was assessed in plasma by ELISA. Briefly, plates were coated with rh-α-GalA (#6146-GH, R&D, Systems), which binds to IgG antibodies against the α-GalA recombinant enzyme (diluted 1:50 in PBS with 5% low fat milk). The signal was detected through HRP-conjugated anti-human IgG H&L secondary antibody (A0293, Sigma-Aldrich) and TMB substrate (Thermofischer #34029). IgG concentration was calculated relative to a standard curve obtained using anti-α-GalA IgG antibody (ab169315, Abcam, Cambridge, UK) and HRP-conjugated anti-mouse IgG H&L secondary antibody (97040, Abcam). Absorbance was measured at 450 nm using a FluorStar Omega plate reader and anti α-GalA IgG antibody concentration expressed as μg/mL.

### 4.6. Statistical Analysis

The statistical analysis was performed using GraphPad Prism v9.1 software (GraphPad Software, Inc., La Jolla, CA, USA). The differences between FD patients and controls, as well as between untreated FD patients (naïve) and those undergoing enzyme replacement therapy (ERT), were evaluated using one-way ANOVA, a non-parametric test, and Kruskal–Wallis multiple comparisons. Dot plots were created to provide a visual representation of the data. Significance levels are denoted as follows: * for *p* < 0.05, ** for *p* < 0.01, and *** for *p* < 0.001, with corresponding *p*-values reported wherever statistically significant. For statistical analysis of longitudinal data, we used a non-parametric Friedman test for the evaluation of each group individually (Ti, 6 months, and 12 months). Subsequently, we performed the Wilcoxon matched-pairs signed-rank test for the detailed bivariate analyses. To assess the differences between groups (controls, naïve, and ERT) in each of the time periods, we used the non-parametric Kruskal–Wallis test and the Mann–Whitney test as post hoc tests for the detailed bivariate analyses. The whole process of the statistical analysis is described in the Appendix A.

Correlation between variables in each group was assessed by calculating Spearman’s correlation coefficients. The significance level of the Spearman coefficient was established for * *p* < 0.05. These tests were performed using SPSS (Statistical Package for the Social Sciences) v.19.

## 5. Conclusions

FD treatments are available and have been demonstrated to meliorate the quality of life of patients, although the currently used biomarkers are not really effective at following-up the disease progression and efficacy of these treatments. In this study, we showed that inflammatory and endothelial dysfunction markers (ADAMTS-13, TNF-α, GDF-15, VEGFA, MPO, and MCP-1) are increased in FD patients’ plasma samples, and can help to evaluate, from a holistic perspective, the risk of disease progression in FD, as well as the response to therapeutic interventions. Therefore, even though the analyzed markers cannot be considered to be accurate biomarkers due to their non-specificity, taken together they provide a signature of useful reference molecules with prognostic value for FD.

## Figures and Tables

**Figure 1 ijms-25-06024-f001:**
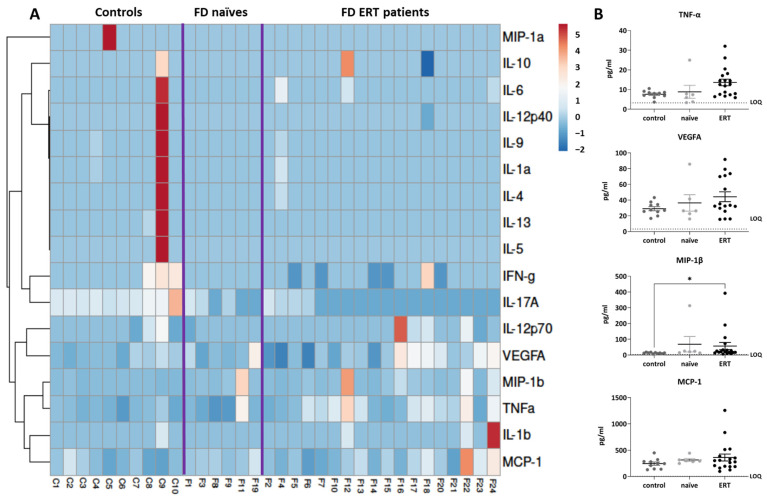
Plasma biomarkers related to inflammatory response were assessed by Luminex-multiplex-ELISA. (**A**) Heat map of assessed biomarkers levels representing Z-score of cytokine concentrations in each sub-cohort, using a color grade scale from dark blue (minimum) to dark red (maximum). Purple lines represent separation between groups. Correlation groups for each biomarker are indicated. (**B**) Dot plots indicating plasmatic concentrations (pg/mL) of the cytokines selected for validation (MIP-1β, MCP-1, VEGFA, and TNF-α) in each subject of the analyzed cohorts (FD non-treated patients: naïve; FD patients treated with ERT: ERT and healthy subjects: controls). Horizontal dotted line indicates assay lower limit of quantification (LOQ). Statistical significance was assessed with one-way ANOVA non-parametric test (Kruskal–Wallis multiple comparisons, * *p* < 0.05).

**Figure 2 ijms-25-06024-f002:**
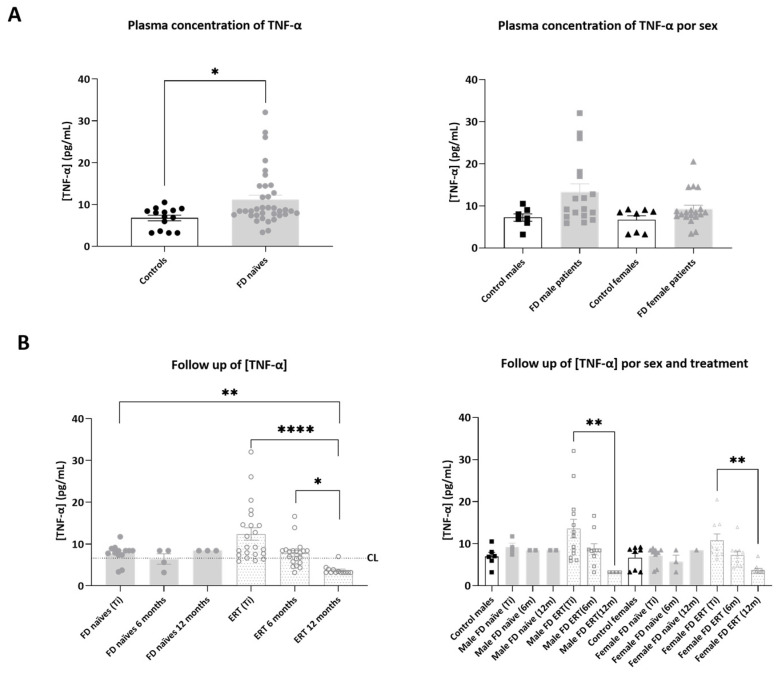
Representation of plasmatic concentration of TNF-α in the analyzed cohorts. (**A**) Left graph shows histograms representing mean concentration (mean ± SEM) of TNF-α (pg/mL) in FD patients (treated or not with ERT) versus healthy controls. The right diagram represents concentrations of the biomarkers in patients and controls divided by sex. (**B**) Histograms in the left panel represent the mean concentration (mean ± SEM) of TNF-α in FD naïve and treated patients at different time points (study onset (Ti), 6 and 12 months after Ti). Dotted line CL represents the mean plasmatic concentration in control subjects. In the right panel, evolution of mean concentration of the biomarker in the treatment groups is discriminated by sex. Statistical significance was assessed with one-way ANOVA non-parametric test (Mann–Whitney or Kruskal–Wallis comparisons, * *p* < 0.05, ** *p* < 0.01, **** *p* < 0.0001).

**Figure 3 ijms-25-06024-f003:**
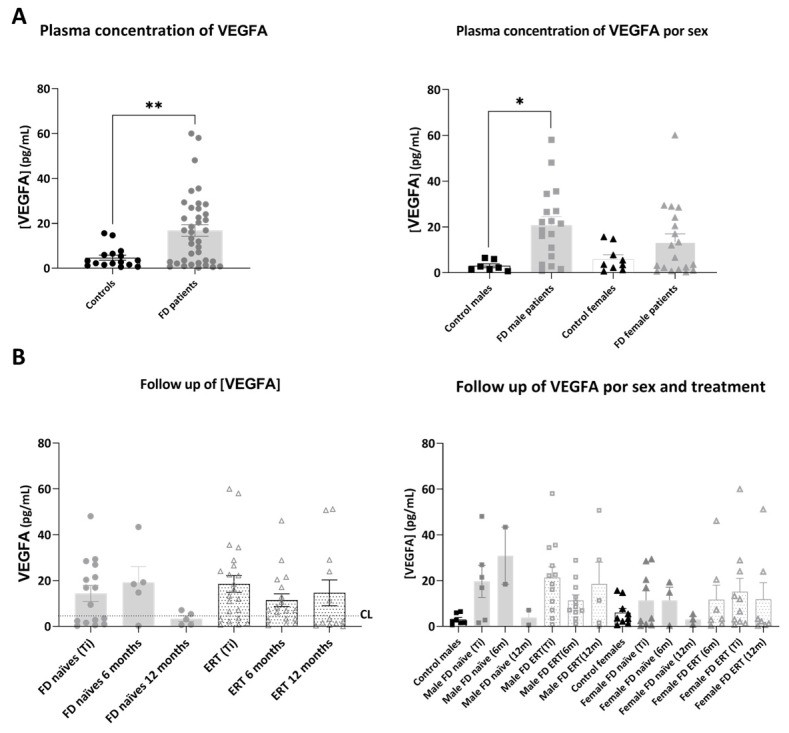
Representation of plasmatic concentration of VEGFA in the analyzed cohorts. (**A**) Left graph shows histograms representing mean concentration (mean ± SEM) of VEGFA (pg/mL) in FD patients (treated or not with ERT) versus healthy controls. The right diagram represents concentrations of the biomarkers in patients and controls divided by sex. (**B**) Histograms in the left panel represent the mean concentration (mean ± SEM) of VEGFA in FD naïve and treated patients at different time points (study onset (Ti), 6 and 12 months after Ti). Dotted line CL represents the mean plasmatic concentration in control subjects. In the right panel, evolution of mean concentration of the biomarker in the treatment groups is discriminated by sex. Statistical significance was assessed with one-way ANOVA non-parametric test (Mann–Whitney or Kruskal–Wallis comparisons, * *p* < 0.05, ** *p* < 0.01).

**Figure 4 ijms-25-06024-f004:**
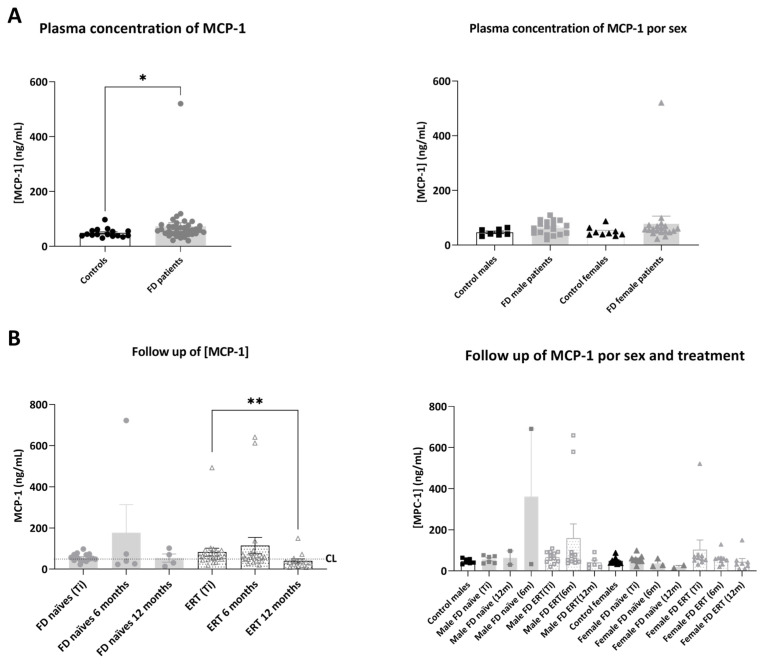
Representation of plasmatic concentration of MCP-1 in the analyzed cohorts. (**A**) Left graph shows histograms representing mean concentration (mean ± SEM) of MCP-1 (ng/mL) in FD patients (treated or not with ERT) versus healthy controls. The right diagram represents concentrations of the biomarkers in patients and controls divided by sex. (**B**) Histograms in the left panel represent the mean concentration (mean ± SEM) of MCP-1 in FD naïve and treated patients at different time points (study onset (Ti), 6 and 12 months after Ti). Dotted line CL represents the mean plasmatic concentration in control subjects. In the right panel, evolution of mean concentration of the biomarker in the treatment groups is discriminated by sex. Statistical significance was assessed with one-way ANOVA non-parametric test (Mann–Whitney or Kruskal–Wallis comparisons, * *p* < 0.05, ** *p* < 0.01).

**Figure 5 ijms-25-06024-f005:**
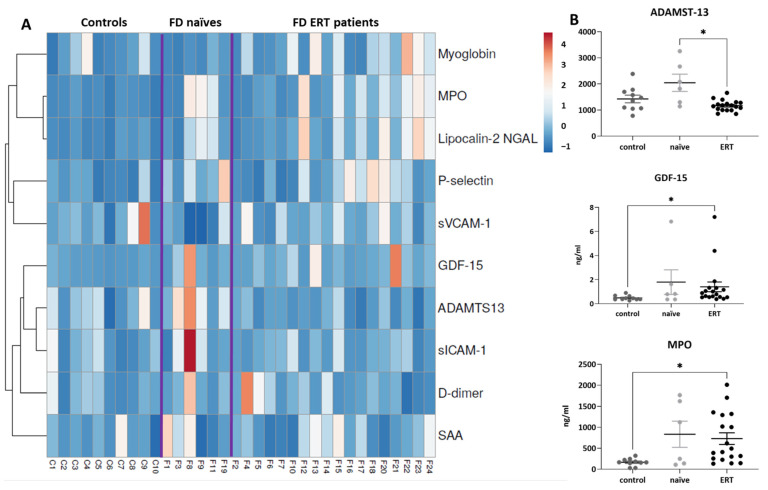
Plasma biomarkers related to cardiovascular risk factors assessed by ELISA-Luminex. (**A**) Heat map of assessed biomarkers levels indicating Z-score of concentrations in a color grade scale from dark blue (minimum) to dark red (maximum) in each sub-cohort as indicated; purple lines represent separation between groups. Correlations among biomarkers are highlighted. (**B**) Dot plots indicating plasmatic concentrations (ng/mL) of the analyzed cardiovascular biomarkers (ADAMTS-13, GDF-15, MPO) in FD non-treated patients (naïve), those treated with ERT, and healthy subjects (controls). Statistical significance was assessed with one-way ANOVA non-parametric test (Kruskal–Wallis multiple comparisons, * *p* < 0.05).

**Figure 6 ijms-25-06024-f006:**
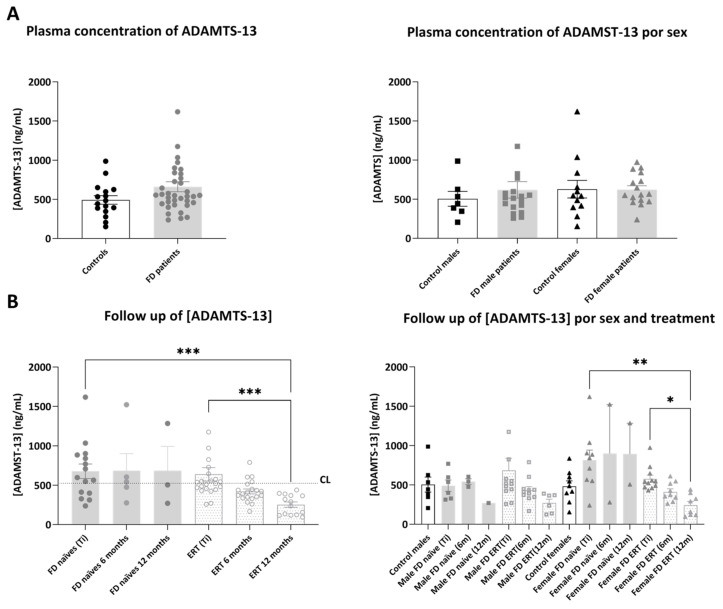
Representation of plasmatic concentration of ADAMTS-13 in the analyzed cohorts. (**A**) Left graph shows histograms representing mean concentration (mean ± SEM) of ADAMTS-13 (pg/mL) in FD patients (treated or not with ERT) versus healthy controls. The right diagram represents concentrations of the biomarkers in patients and controls divided by sex. (**B**) Histograms in the left panel represent the mean concentration (mean ± SEM) of ADAMTS-13 in naïve and treated patients at different time points (study onset (Ti), 6 and 12 months after Ti). Dotted line CL represents the mean plasmatic concentration in control subjects. In the right panel, evolution of the biomarker mean concentration in the treatment groups is discriminated by sex. Statistical significance was assessed with one-way ANOVA non-parametric test (Mann–Whitney or Kruskal–Wallis comparisons, * *p* < 0.05, ** *p* < 0.01, *** *p* < 0.001).

**Figure 7 ijms-25-06024-f007:**
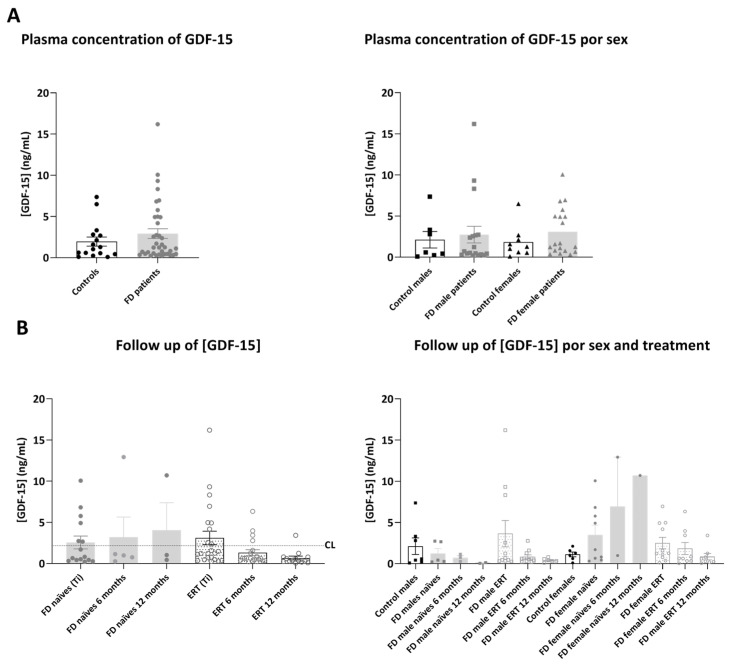
Representation of plasmatic concentration of GDF-15 in the analyzed cohorts. (**A**) Left graph shows histograms representing mean concentration (mean ± SEM) of GDF-15 (ng/mL) in FD patients (treated or not with ERT) versus healthy controls. The right diagram represents concentrations of the biomarkers in patients and controls divided by sex. (**B**) Histograms in the left panel represent the mean concentration (mean ± SEM) of GDF-15 in naïve and treated patients at different time points (study onset (Ti), 6 and 12 months after Ti). Dotted line CL represents the mean plasmatic concentration in control subjects. In the right panel, evolution of the biomarker mean concentration in the treatment groups is discriminated by sex. Statistical significance was assessed with one-way ANOVA non-parametric test (Mann–Whitney or Kruskal–Wallis comparisons).

**Figure 8 ijms-25-06024-f008:**
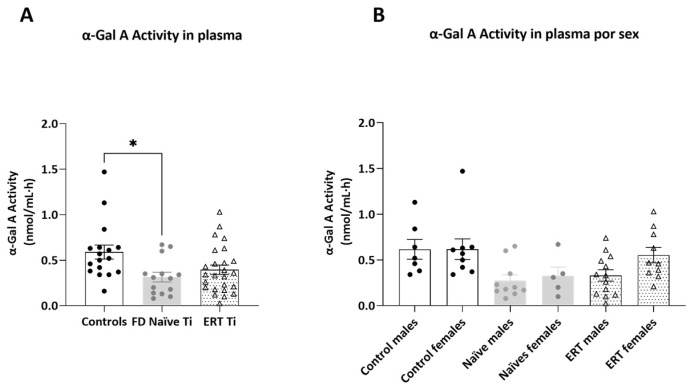
Plasmatic activity levels. Histograms n panel (**A**) represent plasmatic α-GalA activity ± SEM (nmol/mL·h) in controls (white bars), FD naïve patients (light gray bars), and FD patients treated with ERT (white bars with dots). In panel (**B**), α-GalA activity values are separated according to sex. Significance was assessed by one-way ANOVA non-parametric test, Kruskal–Wallis multiple comparisons (* *p* < 0.05).

**Figure 9 ijms-25-06024-f009:**
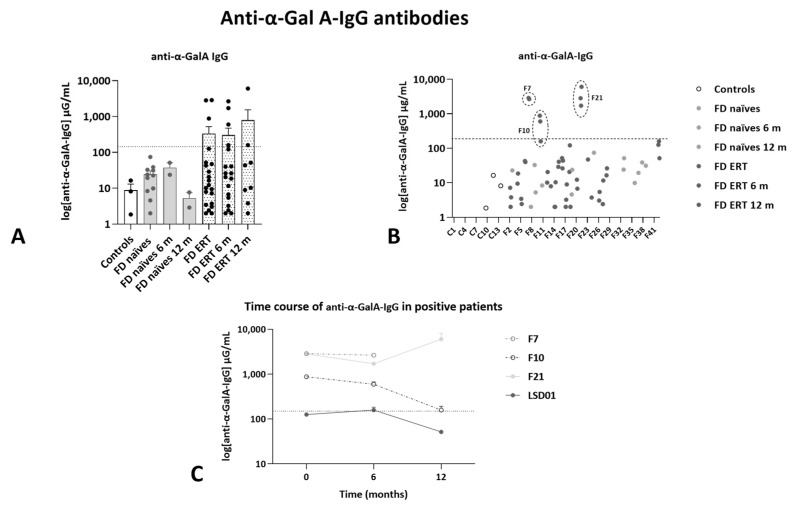
Anti-α-GalA-IgG antibody concentration. Concentration of IgG antibodies determined by ELISA in plasma samples of control individuals and patients in treatment or not with ERT, collected at Ti, T6, T12 months. (**A**) Average values (mean ± SEM) for each group of patients at each collection time. (**B**) Individual values (mean of two replicates) for each subject of the study at all assessed times. (**C**) Time course of anti-α-GalA antibodies for the four patients in panel (**B**), who presented antibody levels close or above the threshold. In all graphs, dotted lines represent threshold.

**Figure 10 ijms-25-06024-f010:**
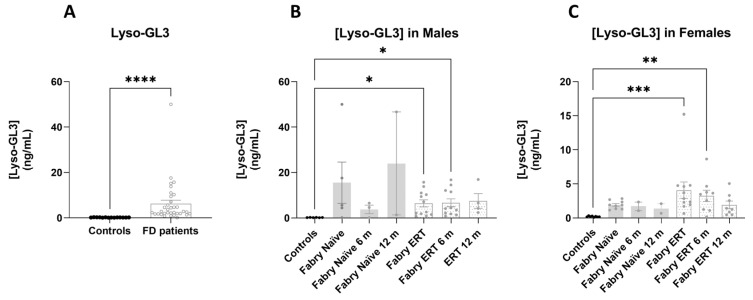
Lyso-GL3 concentration in plasma. Lyso-GL3 concentration in plasma determined by LC-MS in samples from control individuals and patients in treatment or not with ERT (**A**) Average values (mean ± SEM) in FD patients and healthy controls. (**B**) Average values (mean ± SEM) for each group in male subjects over 12 months of study. (**C**) Average values (mean ± SEM) for each group in female subjects. Significance was assessed by one-way ANOVA non-parametric test, Kruskal–Wallis multiple comparisons (* *p* < 0.05; ** *p* < 0.01; *** *p* < 0.001; **** *p* < 0.0001).

**Table 1 ijms-25-06024-t001:** FD patients’ cohort description. Activity of α-GalA (μmol/hL) was measured in dried blood spots (DBSs) at enrollment in the study. NT indicates non-treated FD patient. ERT = enzyme replacement therapy, PC = pharmacological chaperone. N.D. = not available, N.A. = not applicable.

Patient ID	Age	Sex	Activity in DBS (μmol/hL)	Mutation	Treatment	Age of Treatment Onset	Duration of ERT before Enrolment (Years)
F1	33	F	1.35 ± 0.21	p.Pro205Ser	NT ^1^	33 (ERT)	0
F2	51	F	4.80 ± 0.62	p.Gln279Arg	ERT	48	3
F3	26	F	4.89 ± 0.30	p.Gln279Arg	NT	N.A.	0
F4	46	F	3.86 ± 0.13	p.Gln279Arg	ERT	32	14
F5	62	F	5.63 ± 0.03	p.Gln279Arg	ERT	49	13
F6	33	F	4.51 ± 0.54	p.Gln279Arg	ERT	22	11
F7	42	F	3.30 ± 1.26	p.Gln279Arg	ERT	26	16
F8	47	F	1.69 ± 0.03	c.596del	NT	N.A.	0
F9	36	F	2.08 ± 0.77	c.596del	NT	N.A.	0
F10	39	M	1.73 ± 0.07	p.Gln279Arg	ERT	33	6
F11	30	M	1.64 ± 0.70	p.Pro205Ser	NT	N.A.	0
F12	38	M	3.54 ± 0.73	p.Pro205Ser	ERT	N.D.	N.D.
F13	60	F	4.02 ± 0.63	p.Gln279Arg	ERT	51	9
F14	48	F	5.42 ± 2.07	p.Gln279Arg	ERT	45	3
F15	51	M	2.80 ± 0.19	p.Gln279Arg	ERT	48	3
F16	41	F	4.83 ± 0.95	p.Gln250Pro	ERT	37	4
F17	64	F	3.01 ± 0.68	p.Gln250Pro	ERT	60	4
F18	52	M	2.96 ± 1.05	p.Met290Thr	ERT	48	4
F19	34	F	2.51 ± 0.89	p.Gln250Pro	NT	N.A.	0
F20	47	M	2.77 ± 0.62	p.Met290Thr	ERT	36	11
F21	45	M	2.12 ± 0.14	p.Gln250Pro	ERT	41	4
F22	45	M	3.28 ± 0.43	p.Gln250Pro	ERT	41	4
F23	41	M	2.77 ± 0.25	p.Pro205Ser	ERT	N.D.	N.D.
F24	60	F	1.45 ± 0.15	N.D.	ERT	N.D.	N.D.
F25	7	M	1.98 ± 0.78	c.431del	NT	N.A.	0
F26	66	M	3.33 ± 0.22	p.Ser238Asn	ERT	N.D.	N.D.
F27	56	M	3.25 ± 0.08	p.Ser238Asn	ERT	N.D.	N.D.
F28	44	M	2.42 ± 0.52	p.Ser238Asn	ERT	N.D.	N.D.
F33	51	M	2.15 ± 1.18	p.Met290Ile	NT	N.A.	0
F35	51	M	1.71 ± 1.62	p.M187Ile	NT ^2^	51 (PC)	0
F36	42	F	4.77 ± 0.47	p.Gln279Arg	NT	N.A.	0
F37	51	M	2.58 ± 0.75	p.Ser238Asn	NT	N.A.	0
F38	50	F	5.67 ± 1.53	p.M187Ile	NT	N.A.	0
F39	15	F	4.32 ± 1.55	p.M187Ile	NT	N.A.	0
F40	13	F	2.55 ± 0.35	p.M187Ile	NT	N.A.	0
LSD-01	40	M	3.84 ± 0.09	p.Gln279Arg	ERT	28	12

^1^ At T6 treated with ERT; ^2^ at T12 started PC therapy.

## Data Availability

Data are stored in the repository of Servizo Galego de Saude (regional public healthcare system) and are available upon request to the corresponding author.

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
