# Peer review of "Inflammatory and Cardiovascular Biomarkers to Monitor Fabry Disease Progression"

_ijms, 2024, doi:10.3390/ijms25116024_

Round 1
Reviewer 1 Report
Comments and Suggestions for Authors
I believe that the paper
“Inflammatory and cardiovascular biomarkers to monitor Fabry 2 disease progression”
It is of utmost importance to reiterate the need for a holistic approach in evaluating the effectiveness of treatments for Fabry disease (FD). As the authors correctly point out, the proteins analysed by Ortolano and colleagues cannot be used as stand-alone biomarkers due to their lack of specificity. However, a combination of biomarkers can create a reference signature of molecules that can be used for early diagnosis, disease progression monitoring, and treatment effectiveness evaluation by analysing blood samples.
Having stated that I believe the paper needs improvements, but does not require further eperiments.
General remarks
● It is fundamental to maintain consistency in protein names (e.g. MIP-1beta vs MIP1-beta vs MIP-beta)
● Check punctuation and screen the text for typographical errors.
● The results section is extremely redundant and confusing. Please be more concise and selectively include only the most informative and valuable plots to support the narrative. Remove the less relevant ones or place them in supplementary materials.
● The discussion (459-660) should be shortened and made less redundant relative to the results section.
Introduction
I am citing the paper where it states:“A third option for FD medication was more recently approved and it consists in oral administration of a small molecule, Migalastat, that acts as a pharmacological chaperone (PC), stabilizing the mutated enzyme [17], however this drug is only indicated for patients with specific GLA variants. Migalastat showed to improve LVH index, but cardiac biomarker assessment studies upon chaperone treatment are still poorly available.”
● Why “third”? I have only seen reference to ERT so far, but perhaps the authors should also mention the possibility of substrate reduction therapy and cite appropriate papers such as
Guérard, Nicolas, et al. "Lucerastat, an iminosugar for substrate reduction therapy: tolerability, pharmacodynamics, and pharmacokinetics in patients with Fabry disease on enzyme replacement." Clinical Pharmacology & Therapeutics 103.4 (2018): 703-711.
Deegan, Patrick B., et al. "Venglustat, an orally administered glucosylceramide synthase inhibitor: Assessment over 3 years in adult males with classic Fabry disease in an open-label phase 2 study and its extension study." Molecular genetics and metabolism 138.2 (2023): 106963.
● Indeed, Migalastat is only indicated for patients with specific missense mutations. The large genotypic and phenotypic spectrum should be considered citing appropriate papers such as:
Palaiodimou, Lina, et al. "Fabry disease: current and novel therapeutic strategies. A narrative review." Current Neuropharmacology 21.3 (2023): 440.
Burlina, Alessandro, et al. "An expert consensus on the recommendations for the use of biomarkers in Fabry disease." Molecular Genetics and Metabolism (2023): 107585.
It should be noted that combined therapies, specifically the use of pharmacological chaperones in combination with anti-inflammatory molecules, have been proposed.
Monticelli, Maria, et al. "Drug repositioning for Fabry disease: acetylsalicylic acid potentiates the stabilization of lysosomal alpha-galactosidase by pharmacological chaperones." International journal of molecular sciences 23.9 (2022): 5105.
Monticelli, Maria, et al. "Curcumin Has Beneficial Effects on Lysosomal Alpha-Galactosidase: Potential Implications for the Cure of Fabry Disease." International Journal of Molecular Sciences 24.2 (2023): 1095.
Results
● There is incorrect section numbering: "Results" is Section 2, yet subsections are numbered as 3.x. Please correct this.
● The acronym DBS is not defined in Table 1. Please provide its definition in the caption together with the others (line 146)
● Generally, the support provided by the heatmap for lines 164-171 is very weak. Panel A is extremely difficult to interpret. The scaling of levels by subcohort prevents the comparison of different subcohorts and does not allow for the evaluation of 'increased levels' stated in the text (adding the averages in the caption and/or text could be enough). Furthermore, Figure 1B lacks statistical support and testing.
● Figure 2 (The issues noted in Figure 2 also apply to varying extents to Figures 3, 4, 6, and 7.)
○ Values for Naive vs ERT treated are mentioned in the text (187-189) but are absent from the left panel of the figure.
○ The text does not address the sex stratification shown in the right subplots of panels A and B.
○ Are the 6- and 12-month measures from the same patients?
○ Consider reducing Figure 2 to just Fig 2A left, and Fig 2B left and either moving the rest to supplementary materials or removing them entirely.
○ In Fig 2C, F33 is an outlier that significantly distorts the plot. Please remove it.
○ What contribution does Fig 2C make to the results? Since the time-dependent decrease is already discussed in Fig 2B, consider keeping panel C as supplementary material (and adjust the text accordingly).
● There is a critical misunderstanding about the role of STRING (lines 447-454). STRING is NOT a simulator. Rather, it is a resource that provides Protein-Protein Interaction Networks based on Functional Enrichment Analysis, so that should read, " We conducted an in-silico analysis using STRING software to explore evidence-based protein-protein interactions between the identified biomarkers and α-Gal A."
○ To increase robustness, I would either remove text-mining (string setting) OR change from evidence to confidence. The network does not have a legend.
○ The link between A4GALT and ITGA2B (lines ) is exclusively by text mining without proof of direct relation (they are linked because they are reported to be both involved in glycosylation), making the evidence feeble. I suggest removing the entire analysis.
○ Add a caption to Figure 11 explaining the colour code for edges (i.e. segments joining proteins).
○ Remove the word "modelling" from the caption of Figure 11 (line 457)
Reviewer 2 Report
Comments and Suggestions for Authors
Alonso-Núñez et al. have studied longitudinally inflammatory biomarkers in a relatively small group of patients with Fabry disease and controls. Some patients were ERT naive and some already on ERT. For the most part, this is a confirmatory study that does not add much to our knowledge. Overall, the lack of uniformity in a small group (some on ERT some not and no pre-ERT baseline) prevents reaching any conclusions about these data. Further comments:
1. Statistical aspects: The data need an evaluation by a statistician to assess the need to correct the analysis for multiple comparisons beyond Kruskal-Wallis multiple comparisons. Also, please explain how you deal with outliers that are present in a number of cases (e.g. TNFalfa Fabry naive). The data need to separate males from females more clearly in the results and the graphs.
2. TNFalfa: the level on ERT seem higher thank in ERT naive. This suggest that the elevation may be related to the inflammatory/immune response to ERT. This suggests that there is no Fabry-related elevation. Same thing probably with GDF-15.
3. Besides the inflammatory effect of ERT, the authors to not associate the data to age, sex, and various health conditions such as eGFR and cardiac status. This is important because a lot of these abnormalities are secondary (no less important) to organ dysfunction and not to the primary disease process. This is important in order to understand why a biomarker is elevated in some patients but not in others. Also, one should be very careful to ascribe a "positive ERT effect" to any of those data. ERT has no effect on VEGF and "improvement" on ERT in some cases likely reflects decrease immune response to the infused enzyme.
4. The authors need to do a better job with their reference citations. For example, when they discuss MPO (they should define it as myeloperoxidase in the text) they should cite: Kaneski et al. Neurology, 2006 Dec 12;67(11):2045-7..VEGF elevation was first described in children in the study Moore DF et al. Proc Natl Acad Sci U S A. 2007 Feb 20;104(8):2873-8. The reference regarding inflammatory abnormalities in Fabry disease should also be cited. All this emphasize the at best confirmatory nature of this manuscript.
Round 2
Reviewer 2 Report
Comments and Suggestions for Authors
The authors responded positively to my comments. I have no new comments.
Comments on the Quality of English LanguageI think it is generally OK.